# Chromosomal Instability-Driven Cancer Progression: Interplay with the Tumour Microenvironment and Therapeutic Strategies

**DOI:** 10.3390/cells12232712

**Published:** 2023-11-26

**Authors:** Siqi Zheng, Erika Guerrero-Haughton, Floris Foijer

**Affiliations:** 1European Research Institute for the Biology of Ageing (ERIBA), University Groningen, University Medical Center Groningen, 9713 AV Groningen, The Netherlands; s.zheng01@umcg.nl (S.Z.); eguerrero@gorgas.gob.pa (E.G.-H.); 2Department of Research in Sexual and Reproductive Health, Gorgas Memorial Institute for Health Studies, Panama City 0816-02593, Panama; 3Sistema Nacional de Investigación, SENACYT, Panama City 0816-02593, Panama

**Keywords:** chromosomal instability, tumour microenvironment, extracellular vesicles, cancer therapy, immune modulation, metabolic vulnerabilities, extracellular matrix, metastasis

## Abstract

Chromosomal instability (CIN) is a prevalent characteristic of solid tumours and haematological malignancies. CIN results in an increased frequency of chromosome mis-segregation events, thus yielding numerical and structural copy number alterations, a state also known as aneuploidy. CIN is associated with increased chances of tumour recurrence, metastasis, and acquisition of resistance to therapeutic interventions, and this is a dismal prognosis. In this review, we delve into the interplay between CIN and cancer, with a focus on its impact on the tumour microenvironment—a driving force behind metastasis. We discuss the potential therapeutic avenues that have resulted from these insights and underscore their crucial role in shaping innovative strategies for cancer treatment.

## 1. Chromosomal Instability (CIN) in Cancer

Chromosomal instability (CIN), a process driving genomic alterations [1], promotes cancer cells to adapt to stress [2,3]. CIN refers to an increased frequency of errors in mitosis and can lead to alterations of whole chromosome copy numbers as well as to structural rearrangements [1,2]. The result of CIN is aneuploidy, a state in which cells have an unbalanced DNA content. Furthermore, CIN will lead to variability between cancer cell karyotypes, a concept known as intratumour karyotype heterogeneity. CIN is prevalent during tumour development, and its severity correlates with tumour development stage, genomic heterogeneity, immune escape, and metastasis rates [4,5,6,7,8]. However, we still do not fully understand the relationship between CIN and cancer. While CIN has the potential to promote tumourigenesis, CIN and the resulting aneuploidy landscapes differ between cancer types [3,9,10,11]. 

CIN in cancer cells is caused by various, often co-occurring factors, each yielding unique CIN signatures [12,13]. For instance, replication stress, i.e., the slowing or stalling of DNA replication, can lead to unique structural DNA copy number variations, but also large-scale chromosome losses [9]. Transient anomalies in mitotic spindle geometry promote chromosome segregation errors, thus leading to whole chromosome copy number changes [14]. Similarly, disruptions to the kinetics of microtubule polymerization and depolymerization will affect microtubule-kinetochore interactions and increase the chance of chromosome mis-segregation, thus promoting whole chromosome copy number alterations [15]. Telomere attrition can lead to CIN via breakage–fusion–bridge cycles of chromosomes with shortened telomeres, which will lead to chromosome fragmentation and thus structural copy number changes [16]. Collectively, a variety of factors, including but not limited to deregulated kinetochore-microtubule interactions, replication stress, and telomere erosion, thus contribute to the diverse karyotypic alterations instigated by CIN [9,14,15,16,17,18]. 

While novel cancer therapies have significantly improved survival rates and quality of life in patients with various malignancies, therapy resistance is one of the most important factors affecting overall patient survival [19,20]. The emergence of drug resistance is often attributed to new mutations in tumour cells during treatment [21,22,23]. CIN is an important contributor to such drug resistance [24,25]. For instance, delays in the G1 phase of the cell cycle imposed by aneuploidy, the result of CIN, were described to play a role in increased resistance to chemotherapy drugs like cisplatin and paclitaxel [26]. In line with this, high aneuploidy rates and intratumour karyotype heterogeneity as a proxy of ongoing CIN are associated with higher recurrence rates and worse prognosis in neuroendocrine neoplasms [27,28] and are a crucial indicator of metastasis and therapy resistance in cancers like hepatocellular carcinoma, colorectal cancer, and breast cancer, highlighting its potential as a target for treatment [4,29,30,31].

In this review, we will discuss the relationship between CIN, metastasis, and the TME, and highlight recent work that might provide new directions for therapeutic interventions of CIN^+^ tumours.

## 2. Molecular Mechanisms Underlying Metastasis in CIN^+^ Cancers

As CIN drives karyotype heterogeneity, it contributes significantly to tumour evolution and adaptation [5,24]. While CIN has mostly been considered a process that promotes genetic diversification, recent work reveals a broader impact on tumour progression [7,24]. For instance, the effects of CIN on the tumour microenvironment (TME) and metastatic potential were recently identified as important contributors to the outcome of CIN in cancer [3]. By activating pro-tumour inflammatory signalling pathways, CIN promotes an inflammatory microenvironment that supports tumour growth and cancer cell survival [4,32]. Furthermore, metabolic alterations instigated by CIN lead to TME remodelling, providing tumour cells with nutrients while at the same time suppressing anti-tumour immune responses [3]. These combined effects promote tumour progression and establish a TME that better supports metastasis.

While the effects that CIN has on the TME will likely influence the efficacy of metastatic seeding, CIN also has effects that more directly promote metastatic potential [4,8]. Genetic heterogeneity resulting from CIN helps cancer cells acquire characteristics that increase their invasive capabilities, enhance survival in circulation, and facilitate seeding at distant sites [4,8]. Furthermore, the genetic chaos resulting from CIN triggers the activation of oncogenic pathways, which can directly support metastasis as these pathways frequently intersect with fundamental cellular processes such as metabolic signalling, epithelial-mesenchymal transition (EMT), and immune evasion—all crucial factors for metastasis [3].

One pathway that has increasingly been associated with CIN, the TME, and metastasis is the cGAS-STING pathway. In the next section, we will explain this relationship in more detail. 

### 2.1. cGAS-STING Signalling and Immune Modulation

Under normal conditions, DNA is strictly confined to the nucleus. This prevents DNA from being recognised by the cyclic GMP-AMP synthase (cGAS), which will activate the cGAS-STING signalling pathway, ultimately mounting an innate immune response [33,34,35]. cGAS can be activated by cytoplasmic DNA for instance, DNA exposed from ruptured micronuclei, caused by mis-segregated chromosomes [4,31]. In mammals, cGAS functions as a primary sensor of cytosolic double-stranded DNA (dsDNA) [34,35]. Activation of cGAS leads to the production of cyclic GMP-AMP (cGAMP), which engages the stimulator of interferon genes (STING) [34,35]. This cascade promotes the induction of interferons and various proinflammatory molecules [36,37]. Indeed, the cGAS-STING pathway is upregulated in cells displaying increased levels of CIN [10], thus linking CIN to a primary immune response [4,38]. Persistent DNA-leakage-mediated activation of cGAS-STING may have effects both on the cancer cells that display CIN as well as the immune cells that infiltrated the tumour, as cGAS will enhance the inflammatory response of the cancer cells [4,8]. This will mobilise the immune system and at the same time activate STING in immune cells, thus further activating the immune response [4,8].

Various genes, including KIF2B/C, PICH, MASTL, PRC1, APOBEC3A, and CCAT2 lncRNA, have been shown to modulate tumour immunity through their interactions with the cGAS-STING pathway [4,39,40,41,42,43]. Furthermore, factors like the ectonucleotidase ENPP1 were found to promote metastasis by influencing STING substrate cGAMP, further underscoring the complexity of these interactions [44]. Intriguingly, the cGAS/STING-mediated inflammatory response also appears to provide a targetable vulnerability of cancer cells with induced CIN, as cGAS-STING promotes IL-6-STAT3 pro-survival signalling, presumably for cells to cope with the initial stress response that is imposed by CIN [31]. As such, the interaction between CIN-associated DNA leakage and the activation of cGAS-STING signalling emerges as an important driver of metastasis, with immune components within the TME playing a pivotal role in orchestrating this process. Therefore, a better understanding of this CIN-induced inflammatory response and its effect on the TME will likely reveal novel strategies to exploit this response to treat cancer [3] (also see Figure 1). 

In addition to activating cGAS-STING signalling, CIN also triggers type I interferon (IFN) signalling, a cornerstone of the antiviral defence and tumour immune surveillance [4]. Intriguingly, CIN^+^ cancers were found to circumvent immune surveillance through the amplification of oncogenes, which yielded inhibition of the cGAS-STING pathway and reduced production of type I IFN [4,8]. Additionally, CIN was found to disrupt IFN signalling by altering the expression of essential molecules, such as STAT1 or IRF9, or the amplification of negative feedback mechanisms, such as upregulation of SOCS [4,45]. Therefore, while the cytoplasmic DNA resulting from CIN might trigger cGAS-STING and IFN signalling to activate immune surveillance, CIN^+^ cancers quickly adapt by alleviating the immune-activating IFN signalling to prevent immune clearance. Therefore, new therapies that reactivate immune recognition of CIN^+^ cancer cells are urgently needed. 

### 2.2. Epithelial-Mesenchymal Transition (EMT) and CIN

Epithelial-mesenchymal transition (EMT) is a dynamic cellular process orchestrated by EMT-activating transcription factors (EMT-TFs) like SNAIL, TWIST, and ZEB families, leading epithelial cells to acquire mesenchymal traits [46,47,48]. EMT plays a crucial role in cancer progression by inducing cell polarity and adhesion loss, providing migratory attributes, and imposing a mesenchymal phenotype upon cancer cells [47]. CIN can drive transcriptional changes that induce a transition from a hypermetabolic to a mesenchymal state, a precursor state to metastasis [4,49,50]. The interplay between CIN and EMT is further exemplified in ovarian cancers, in which loss of intercellular junction (IJ) proteins triggers EMT [51]. In this case, CIN leads to copy number changes of chromosomal regions harbouring IJ protein regulators, which will strengthen the EMT phenotype, thus favouring metastatic colonization [51]. 

While CIN can promote EMT, EMT inducers like Twist1 might also induce genomic instability, for instance as shown for colorectal cancer [52,53]. Similarly, disruptions in MASTL kinase activity promote EMT by modulating cell-cell junctions, leading to loss of contact inhibition, but are also associated with increased rates of CIN [40,41]. By promoting karyotype evolution, CIN acts as a catalyst to acquire malignant phenotypes, including EMT [54,55]. For instance, CIN was described to affect cellular pathways like TGF-β and WNT, and can change the tumour microenvironment to promote EMT, for example, through induction of hypoxia [56,57,58]. Additionally, CIN was found to inhibit the expression of the epithelial marker E-cadherin, thus promoting a transition to a more mesenchymal phenotype [59]. 

Collectively, these findings underscore the relationship between EMT and CIN in promoting metastasis. Although we do not yet fully understand this relationship, assessing markers for EMT in combination with inference or quantification of intratumour karyotype heterogeneity as a proxy of ongoing CIN might help predict treatment stratification and outcome [49,57] (also see Figure 1). 

### 2.3. CIN and Metabolic Signalling 

CIN and aneuploidy often result in a deregulation of metabolic signalling, thereby disrupting metabolic homeostasis [60,61,62,63]. Altered metabolic signalling is strongly associated with oncogenesis, particularly in tumours marked by hypoxia, glycolysis, and altered levels of oncogenic metabolites [64,65]. Indeed, a dysregulated metabolism is a hallmark of various cancers, especially those with CIN, fuelling their increased energy demand and cell growth [66]. The relationship between CIN and cellular metabolism is exemplified in oral squamous cell carcinoma (OSCC), in which genetic aberrations in both normal and neoplastic cells promote metastasis through deregulation of metabolic pathways [67]. Similarly, in Ewing sarcoma (ES), CIN-induced hypoxia triggers pathways that instigate further genomic alterations, bone dissemination, and drug resistance [68]. Vice versa, hypoxia, low levels of glucose, and lactic acidosis can induce or exacerbate CIN phenotypes in cancer cells [65,69,70]. Furthermore, environmental stresses like hyperthermia and hypoxia induce stress responses in cancer cells that lead to mitotic defects and karyotypic instability, for instance in colorectal cancer [69]. 

How does CIN lead to this altered cellular metabolism? One important factor is that the resulting aneuploidy will affect copy numbers of metabolic enzymes and thus alter the expression of these genes, resulting in imbalances in various metabolic pathways [62,71]. Additionally, CIN-instigated activation of oncogenes like c-Myc or the loss of tumour suppressors such as p53 were found to reprogram metabolic dynamics [72,73]. By altering expression of genes that sense nutrients, CIN can also impact responsiveness to these nutrients, and will likely change metabolic feedback mechanisms [74]. Furthermore, oxidative stress resulting from CIN can impair mitochondrial function, which will also affect cellular metabolism [75,76]. Finally, CIN may affect copy numbers of genes involved in growth factor signalling, such as insulin or IGF-1 signalling, which are important regulators of the cellular metabolism [77]. Together, these factors explain the profound effect that CIN has on the cellular metabolism [66,78]. 

In conclusion, metabolic rewiring and CIN go hand in hand to drive cancer development, therapy resistance, and metastasis (also see Figure 1). Therefore, a better understanding of these interactions may reveal novel therapeutic avenues to treat cancers with CIN.

### 2.4. CIN and Remodelling of the Micro-Environment

The tumour microenvironment (TME) is a complex environment in which tumour cells originate and reside [79]. It comprises diverse cellular and non-cellular components, including immune cells, stromal cells, glial cells, microvasculature, and various biomolecules [79,80]. Unlike the balanced microenvironment of healthy tissues, the TME actively supports the malignant behaviour of tumour cells [81]. Therefore, understanding how CIN affects the TME is crucial. 

CIN-induced mitotic errors lead to a series of changes that reshape the TME, including shifts in the composition of immune cells, reorganisation of the extracellular matrix, alterations in extracellular vesicle communication, changes in angiogenic factors, and shifts in communication between tumour and stromal cells [3,4,82,83]. While these elements collectively contribute to developing a TME that fosters tumour growth and cancer cell dissemination, it will be essential to separate the responses to CIN from the individual cellular and non-cellular components within the TME. 

#### 2.4.1. Cellular Components in the TME

The tumour microenvironment is composed of diverse cellular components, including T cells, B cells, macrophages, and cancer-associated fibroblasts (CAFs) [80]. The cellular composition of the TME is heavily influenced by a CIN phenotype and resulting aneuploidy in cancer cells as a result of the interaction of cells with CIN and the immune system [84,85,86,87]. However, various contrasting effects of more complex aneuploidy landscapes were reported: some studies have observed increased T cell and macrophage infiltration due to increased immunogenicity from tumour cell genetic diversity, while other studies have shown that tumours with CIN foster an immune-suppressive milieu, preventing the primary CIN-induced immune responses [87,88,89,90]. Likely, these contrasting findings relate to the stage of the tumour, with early CIN^+^ tumours recruiting a tumour-suppressive immune landscape and late tumours circumventing this [91,92]. 

Indeed, CIN as well as the resulting aneuploidy have been associated with significantly altered immune landscapes in the TME [84,85,86,87,93]. For instance, highly aneuploid tumours display decreased immune-mediated cytotoxicity, proinflammatory activities within the microenvironment, and suppressed tumour antigen presentation [93]. Furthermore, ongoing CIN and genome doubling were reported to influence the abundance of infiltrating Treg cells and B cells [88]. Conversely, tumours with low aneuploidy and low metastatic potential displayed increased infiltration by Treg cells and B cells compared to tumours with high aneuploidy rates and high metastatic potential [89,90]. Additionally, transcriptome analyses across multiple cancer types with varying aneuploidy rates showed clear differences between the immune landscapes of tumours with lower versus higher aneuploidy rates [93]. Cancer cells that display CIN were also found to promote recruitment of myeloid-derived suppressor cells (MDSCs) to the TME, possibly due to increased secretion of damage-associated molecular patterns and tumour-derived factors [8]. Furthermore, the chronic inflammation driven by CIN^+^ cancer cells may yield a more tumour-suppressive environment, which in turn will contribute to the dysfunction of CD8 T cells [8,85,94]. Finally, the aneuploidies resulting from CIN can alter the expression of important chemokines, such as CCL5 or CXCL9/10, which will also impact the immune cell composition of the TME [38,95,96].

Besides immune cells, cancer-associated fibroblasts (CAFs) also contribute to an immunosuppressive TME through exosome and growth factor secretion, thus promoting tumour growth and metastasis [97]. While CAFs are considered to play an essential role in tumourigenesis, the relationship between CAFs and CIN phenotypes in cancer cells is poorly understood, except for the observation that CAF infiltrates vary between CIN subtypes [98,99,100]. However, it is becoming increasingly clear that the interaction between aneuploid cells and cellular components of the TME, such as immune cells and CAFs, plays a crucial role in shaping the immune landscape within the TME. 

Collectively, these observations indicate that the degree of CIN significantly impacts the composition of immune cell populations within the TME, thereby establishing a nexus between CIN, immune cell infiltration, and their collective impact on the complex landscape of tumourigenesis [89,90]. As such, the cellular composition of the TME has important implications for the immune response to cancer cells, for tumour growth, and for therapeutic strategies. A better understanding of this cellular makeup will likely yield new insights towards more effective cancer treatment strategies (also see Figure 1).

#### 2.4.2. Non-Cellular Components of the Tumour Microenvironment

In addition to cellular elements, CIN might also influence the composition of non-cellular components of the TME [3]. These non-cellular factors include extracellular matrix (ECM) molecules, extracellular vesicles, and various biomolecules [68,83,101]. Jointly, these factors play an important role in shaping the tumour-supportive environment [68,83,101] (also see Figure 1).

##### The Extracellular Matrix (ECM)

The extracellular matrix (ECM) is primarily produced by cancer-associated fibroblasts (CAFs) and plays an essential role in driving tumour growth and metastasis [97,102,103]. Beyond secreting exosomes and growth factors that fuel tumour development, the ECM contributes to an immunosuppressive TME [97]. The ECM contains key components that shape the TME, including proteins, glycoproteins, and proteoglycans that jointly orchestrate cellular function and structure [102,103].

Gene expression profiling of colorectal cancer samples revealed that high aneuploidy scores and microsatellite instability (MSI) correlate with increased expression of the POSTN gene (periostin), a factor known to remodel the extracellular matrix (ECM) [104]. Furthermore, structural aberrations of chromosome 9, a result of CIN, have been associated with a more rigid TME structure, highlighting the relationship of aneuploidies that result from CIN and the ECM in shaping tumour stiffness [104,105].

Vice versa, rigidity of the tumour microenvironment can also facilitate further chromosomal breaks within cancer cells to progress tumour progression [104,105]. For instance, in glioblastoma (GB) cell lines, extra copies of chromosome 7 (Chr7) correlate with higher glioma grades [101]. This phenomenon coincides with reduced expression of EFEMP1, a constituent of the ECM [101]. Indeed, restoring EFEMP1 levels was found to improve mitotic fidelity, suggesting that modulating the ECM by restoring EFEMP1 levels might reduce CIN rates in cancer [101].

Taken together, these observations underscore the importance of understanding the effects of CIN^+^ cancer cells on the ECM [101,104]. As CIN is influenced by the ECM and vice versa, a better understanding of this relationship could yield insights towards targeted therapeutic interventions that exploit this relationship (also see Figure 1).

##### Extracellular Vesicles (EVs)

Extracellular vesicles (EVs) are released by various cell types and act as carriers for RNA, DNA, and proteins, enabling bidirectional communication between cancer cells and the TME to modify signalling cues in recipient cells [106,107]. Tumour-derived EVs, as well as EVs released by non-malignant cells (nmEVs), can contribute to cancer development [107]. For instance, the surface composition of exosomes, a subtype of EVs, can direct exosomes towards specific cell populations and organs, influenced by factors like cytokines, thus promoting organ-specific colonisation by altering the microenvironment [108]. Exosome colonisation will affect integrins in the recipient tissue, ultimately initiating a local inflammatory response and formation of a pre-metastatic niche [109]. Therefore, quantification of the exosome content in blood might aid in predicting metastatic patterns and thus guide clinical decisions [110].

Prior work has revealed many important interactions between CIN and EVs [83,111,112,113]. As an example, amplification of centrosomes, another driver of CIN, triggers increased secretion of small extracellular vesicles (SEVs) as a result of lysosomal dysfunction and increased levels of ROS [88]. In the case of pancreatic cancer, these SEVs can activate pancreatic stellate cells, increasing pancreatic cancer invasiveness and fibrosis [83]. 

What content within the EVs can explain their effect on the TME? Functional characterization of tumour EVs has revealed a three-miRNA signature that might serve as an indicator for genomic instability in breast cancer [113]. As EVs can be exploited as minimally invasive biomarkers for early diagnosis, these findings might have important clinical significance [113]. Similar patterns were found for gastric cancer. Here, RNA signatures identified in EVs that were isolated from 3D cultures correlated with CIN and poor patient survival, further underscoring the potential of EVs in detecting CIN^+^ cancers [112]. While many other factors in EVs will play a role in shaping the TME, collectively, these findings highlight the potential role of EVs as messengers and as promising biomarkers to detect CIN^+^ cancers.

In summary, CIN plays a crucial role in promoting metastasis by affecting both the cellular and non-cellular components of the TME [3]. CIN is associated with an immunosuppressive environment, which is characterised by the presence of myeloid-derived suppressor cells and regulatory T cells that help cancer cells evade detection [8,85,114,115]. CIN also leads to changes in the ECM, making tumours more rigid and invasive [104,105]. Each of these effects, along with CIN’s ability to promote angiogenesis [82], have the potential to be exploited as therapeutic targets to treat cancers with a CIN phenotype (Figure 1).

## 3. Targeting of Mechanisms to Counteract CIN-Driven Cancer

Developing better therapeutic strategies that exploit the effects of CIN in cancer is expected to advance cancer treatment. In this section, we review various approaches to target specific mechanisms influenced by CIN, including immune modulation, metabolic reprogramming, or targeting the extracellular matrix or STING pathway. 

### 3.1. Targeting the CIN-Altered Immune Landscape

The inflammatory response that results from CIN activates the immune system, and this likely explains why highly aneuploid cancers circumvent immune recognition, leading to immune-depleted TMEs [84,85,86,87,93]. Therefore, reinstating a normal inflammatory response might provide a promising strategy to treat CIN^+^ cancers across cancer types [4,8,31,116] (also see Figure 2). However, further work is required to better understand what modulates the inflammatory response imposed by CIN and how CIN^+^ cancer cells circumvent this before this concept can be translated into tangible clinical benefits.

### 3.2. Exploiting Metabolic Vulnerabilities

CIN has major effects on the metabolism of cancer cells, which contributes to their aggressive behaviour [3]. Therefore, targeting the altered metabolic pathways in CIN^+^ cancer cells might provide a powerful means to impair proliferation of these cells [117]. Several approaches target the glucose dependency of CIN^+^ cancer cells, for instance, through modulation of lactate levels [66,117,118]. Such treatment reshapes the TME and makes the cancer cells more dependent on glucose, thus rendering them more vulnerable to glucose deprivation [66,117,118]. 

STING is another factor that links CIN, innate immunity, and the cellular metabolism [3,119]. Recent work revealed that activation of STING involves proton leakage, which impacts cellular processes like autophagy and the immune response [119]. This newly discovered role of STING could mean that targeting the proton-transporting activity of STING might disrupt the potential of cancer cells to adapt their metabolism. Furthermore, other processes downstream of STING, such as autophagy and activation of the inflammasome, might impact the metabolic adaptation that promotes the survival of chromosomally unstable cancer cells [119,120].

While more work is required, jointly this work suggests that CIN leads to metabolic and immune-related vulnerabilities that can be exploited in cancer therapy (also see Figure 2).

### 3.3. Manipulating CIN and Its Downstream Effects

Manipulating CIN rates might provide another powerful strategy to selectively treat CIN^+^ cancers [121,122]. Increasing the rate of CIN can be accomplished by modulating factors that are involved in faithful chromosome segregation, like the centrosomes, microtubule-kinetochore interactions, and the spindle assembly checkpoint [123,124,125]. One such factor is KIF18A, for which inhibition was found to disrupt spindle microtubules, causing mitotic delays and cell death selectively of CIN^+^ cells [126]. Similarly, the CDK2/CDK9 inhibitor CYC065 (Cyclacel) was found to induce mitotic catastrophe in CIN^+^ cancer cells, restraining tumour growth and underscoring its potential to treat cancers that display high-grade aneuploidy and substantial intratumoural karyotype heterogeneity [127]. Furthermore, drug-mediated inhibition of Src1 was found to increase cell death of CIN^+^ cancer cells via the deregulation of microtubule-kinetochore interactions [125].

In conclusion, the strategic manipulation of CIN phenotypes in cancer presents another potential avenue for targeted cancer therapy (also see Figure 2). By targeting the mechanisms underlying CIN and capitalising on its vulnerabilities, CIN^+^ cancer cells can be selectively targeted. While promising first steps have been made, a further systematic exploration of synthetic lethal interactions instigated by CIN is required to fully exploit the potential of this approach. 

### 3.4. Targeting Extracellular Matrix (ECM)

Mitigating CIN phenotypes by targeting the ECM might provide another strategy to exploit unique features of CIN^+^ cancers in therapy [101,105]. To develop such strategies, we first need to better understand how ECM stiffness and mechano-transduction influence DNA repair and chromosome segregation [128,129]. One approach that deserves experimental validation is to target mechano-transduction-associated genes like MAP4K4/6/7 kinases to improve DNA damage repair in CIN^+^ cancers, mitigating CIN in these cancer cells [129]. Similarly, single-cell RNA sequencing of glioblastoma (GB) samples revealed that the ECM-related genes LOX, COL6A2, and TGFB1 are important factors that drive glioma progression, suggesting that targeting these ECM components could be a viable strategy to eradicate cancer stem cells [128]. 

These findings underscore that ECM-targeted strategies might have the potential to manipulate aneuploidy landscapes and the consequences of these across cancers (also see Figure 2). Therefore, therapeutic approaches might emerge in the coming years that disrupt ECM-mediated processes to reduce intratumour karyotype heterogeneity and thus improve treatment outcome.

### 3.5. Targeting Metastasis and STING 

As discussed earlier, CIN promotes metastasis, and STING is a proposed mediator of this effect [4,8]. Activated by micronuclei and chromosome bridges, cGAS-STING signalling promotes both tumour inflammation and progression [130]. Intriguingly, inactivation of key components in this signalling route, such as cGAS, STING, TBK1, and IRF3, also increases micronuclei formation and chromosome mis-segregation, exemplifying the multifaceted role of cGAS-STING signalling in cancer [130]. Furthermore, in addition to its interferon-inducing function, STING also functions as a proton channel [119]. This latter role is particularly relevant for cancer therapy, as inhibiting the STING-induced proton flux might impede noncanonical autophagy and inflammasome activation, presenting another promising avenue to target CIN^+^ cancers [119]. Indeed, the efficacy of selective STING inhibitors to impair CIN-driven metastasis spans diverse cancer types, and their effectiveness correlates with tumour cell-intrinsic STING activity [4,8,130].

The experimental evidence that STING is activated by CIN, in combination with its role in modulating the TME to promote metastasis, underscores the clinical potential of modulating STING activity in cancer. Further work is required to determine when STING should be inhibited to block its tumour-promoting effects, such as driving CIN^+^ cancer cell survival and metastasis [130], or activated to stimulate its tumour-suppressive effects, including immune clearance of cancer cells [131]. 

Altogether, these findings highlight the various effects that CIN has on the immune landscape of cancers. Manipulating the immune composition of the TME of CIN^+^ cancers, exploiting CIN-dependent metabolic vulnerabilities, manipulating CIN rates, modulating of the ECM, and exploiting the multifaceted role of STING signalling all emerge as promising strategies to battle CIN^+^ cancers (Figure 2). However, reaching the full potential of these strategies will require more research on the interplay between CIN, the TME, and cancer progression.

## 4. Conclusions

In this review, we have explored the effects that CIN can have on the TME locally and at distant sites. As CIN discriminates cancer cells from non-cancer cells, targeting CIN provides an attractive target for selective cancer therapy. CIN results in an accumulation of aneuploid cells with various karyotypes, thus promoting cancer cell evolution. As a genome-shaping factor, CIN is associated with tumour recurrence, metastasis, and treatment resistance. Simultaneously, CIN promotes inflammatory phenotypes in cancer cells as a result of chronic cGAS-STING pathway activation, which ultimately leads to mobilization of immune cells to clear CIN^+^ cells. CIN^+^ cancer cells appear to circumvent this immune recognition through various mechanisms. Our growing understanding of this inflammatory response and mechanisms of immune suppression within the TME provide novel opportunities for therapeutic intervention, including immune modulation to reactivate the immune response and clear CIN^+^ cancer cells, or inhibition of the CIN-induced inflammatory response to kill CIN^+^ cancer cells. While the translation to the clinic of these approaches still requires further work, they might provide important next steps towards innovative therapies that target CIN^+^ cancers more effectively with fewer side effects.

## Figures and Tables

**Figure 1 cells-12-02712-f001:**
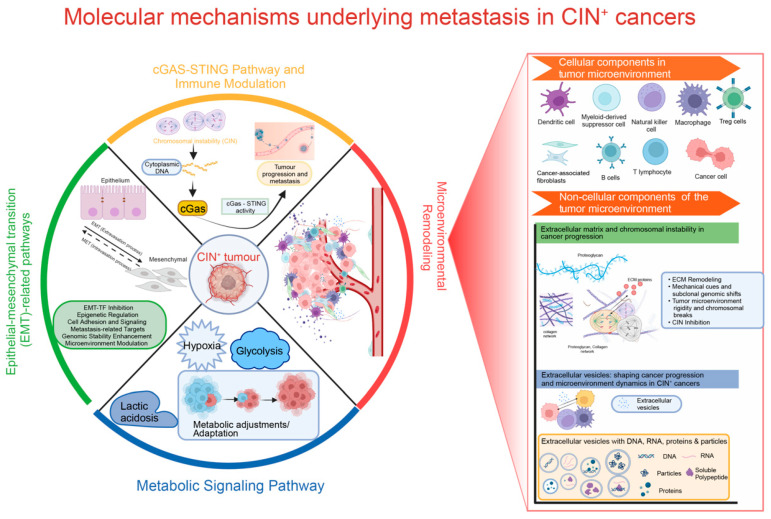
Molecular mechanisms that underly the relationship between CIN and metastasis. cGAS-STING signalling mediates immune modulation in response to DNA leakage, connecting inflammation, genetic diversity, and metastasis, and providing potential therapeutic targets.

**Figure 2 cells-12-02712-f002:**
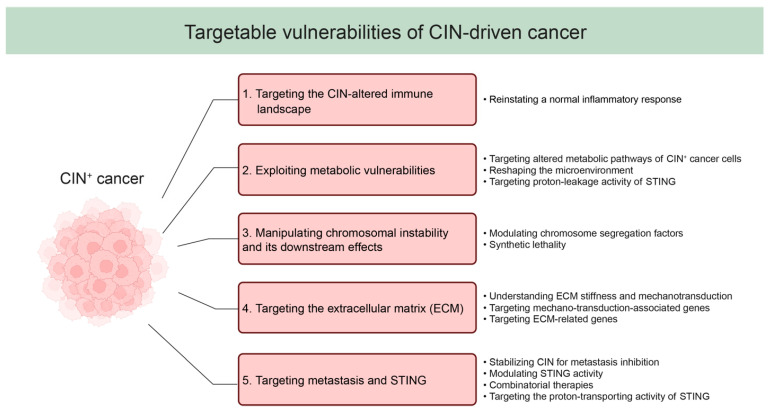
Strategies for targeting chromosomal instability (CIN)-driven cancer. This diagram provides an overview of potential strategies for targeting CIN-driven cancer.

## Data Availability

No new data were created or analyzed in this study. Data sharing is not applicable to this article.

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
