# Peer review of "Chromosomal Instability-Driven Cancer Progression: Interplay with the Tumour Microenvironment and Therapeutic Strategies"

_cells, 2023, doi:10.3390/cells12232712_

Round 1
Reviewer 1 Report
Comments and Suggestions for Authors
This is a very interesting and timely review about the relationship between cancer chromosomal instability (CIN) and the tumour microenvironment (TME). The review is well structured and provides a summary of how CIN is linked to metabolic signalling, the EMT and immune modulation via the cGAS-STING pathway and other immune system components in the TME.
I have only two minor points:
1.) The sentence "CIN refers to an increased rate of changes in chromo-
some numbers or structural copy number alterations during cell division, thus promoting chromosome copy number alterations and unbalanced structural rearrangements [1,5,6]." is a bit tautological. Does it mean CIN is self accelerating?
2.) There are some articles missing. Please read the text again very carefully and include all missing articles.
Otherwise I think, the review paper is great!
Comments on the Quality of English Language
Good english, only some articles missing.
Author Response
Reviewer 1
This is a very interesting and timely review about the relationship between cancer chromosomal instability (CIN) and the tumour microenvironment (TME). The review is well structured and provides a summary of how CIN is linked to metabolic signalling, the EMT and immune modulation via the cGAS-STING pathway and other immune system components in the TME.
I have only two minor points:
1.) The sentence "CIN refers to an increased rate of changes in chromo-
some numbers or structural copy number alterations during cell division, thus promoting chromosome copy number alterations and unbalanced structural rearrangements [1,5,6]." is a bit tautological. Does it mean CIN is self accelerating?
CIN is not inherently self-accelerating; instead, it is characterized by an increased frequency of chromosomal alterations that occur during cell division. This phenomenon can result in a cascade of genetic diversity within a tumour, which may appear to accelerate as the tumour evolves under certain oncogenic stress conditions. However, this acceleration is not inherent to CIN itself, but rather a result of the interaction between CIN and various cellular processes that react to genetic alterations. Therefore, the reviewer’s comments that the previous phrasing was tautological is spot on.
“CIN refers to an increased frequency of errors in mitosis and can lead alterations of whole chromosome copy numbers as well as to structural rearrangements [1,2].”
We hope this addresses your concern and clarifies the concept as intended in our study.
1.2
2.) There are some articles missing. Please read the text again very carefully and include all missing articles.
Thank you for your feedback and for pointing this out. We have thoroughly reviewed the text and incorporated a number of extra references where applicable and also replaced references when they were not making the point that we were citing them for (also see comments reviewer 3).
Reviewer 2 Report
Comments and Suggestions for Authors
In the Review manuscript entitled "Chromosomal Instability-Driven Cancer Progression: Interplay with the Tumour Microenvironment and Therapeutic Strategies," Siqi Zheng, Erika Guerrero-Haughton, and Floris Foijer explore the influence of chromosomal instability on the tumor microenvironment and discuss potential therapeutic approaches for cancer treatment. The review is commendably well-written and organized. However, there are two major issues that need to be addressed:
1. Inappropriate terminology usage:
a) The authors frequently use the term "chromosomal instability (CIN)" when "aneuploidy" or "heterogeneity" would be more appropriate. It's crucial to distinguish between these terms since CIN and aneuploidy represent distinct traits with potentially different clinical implications. The rates of random chromosomal aberrations (CIN) should not be conflated with the level of aneuploidy, as various factors contribute to the selection of chromosomal aberrations leading to clonal aneuploidy.
b) At times, CIN is used as an umbrella term, encompassing the entire CIN phenotype (including aneuploidy, heterogeneity, instability, micronuclei, lagging chromosomes, multipolar mitoses, etc.), while in the next paragraph, it may be used to specifically refer to the instability process itself, measurable by rates of chromosomal instability. Clear delineation of terminology, along with an indication of how specific aspects of the CIN phenotype relate to characteristics of the tumor microenvironment and potential treatment strategies, would enhance the manuscript's clarity.
c) The manuscript refers to "CIN rates" in relation to both cell lines and tumors (e.g., Lines 31, 42, 170...). Please explain how CIN rates were measured for tumor samples in the cited papers.
2. Inadequate citation relevance: The manuscript includes a total of 129 references, some of which are not directly relevant to the statements they are meant to support. In several instances, these references are placed after statements that are either unrelated or only tangentially related to the issues discussed in the cited papers. This issue persists throughout the manuscript and necessitates a thorough review of the cited papers by the authors to ensure that references are appropriately placed or replaced with more relevant ones.
In conclusion, this review offers a substantial contribution to the field of tumor biology and the evolution of cancer cells. To enhance its suitability for publication, the authors must address the two major problems outlined above. Once these issues are rectified, the manuscript should be well-positioned for publication in the journal.
Minor points:
Ln 14-15: sentence needs editing.
Ln 207-208: sentence needs editing.
Comments on the Quality of English Languagegood english
Author Response
Reviewer2
Siqi Zheng et al. reviewed the role of chromosomal instability in cancer progression. This topic is interesting, and the author give insight into multiple consequence of chromosomal instability, and provide some knowledge of finding potential therapeutic strategies. My major concern is it seem that the author mainly focuses on the results in the review, while the relationship with CIN and the mechanism underlying, should be emphasized in the manuscript.
2.1
Although this manuscript is well prepared, the language should be polished by a native speaker at deep extent.
Thank you for your feedback. We have reviewed text once more in detail, ensuring that all spelling and grammatical errors have been fixed. We believe that the revised manuscript now meets the necessary linguistic standards.
2.2
In the second part, the author should at least give a brief introduction and explanation of cause of chromosomal instability in cancer cells, such as karyotypic changes, chromosome mis segregation, and some great reviews could be referred to. (1)
Bakhoum S F, Cantley L C. The multifaceted role of chromosomal instability in cancer and its microenvironment[J]. Cell, 2018, 174(6): 1347-1360.
Thank you for your constructive feedback on the second part of our manuscript. We agree that an introduction and explanation of the causes of chromosomal instability in cancer cells is beneficial for the readers.
We therefore added a brief introduction that describes some of the driving events of chromosomal instability in cancer cells/ Additionally, we refer to more detailed reviews that offer a thorough exploration of this topic. The relevant section is copied hereunder:
“CIN in cancer cells is caused by various, often co-occurring factors, each yielding unique CIN signatures [12,13]. For instance, replication stress, i.e. the slowing or stalling of DNA replication, can lead to unique structural DNA copy number variations, but also large-scale chromosome losses [9]. Transient anomalies in mitotic spindle geometry promote chromosome segregation errors, thus leading to whole chromosome copy number changes [14]. Similarly, disruptions the kinetics of microtubule polymerization and depolymerization will affect microtubule-kinetochore interactions and increase the chance for chromosome missegregation, thus promoting whole chromosome copy number alterations [15]. Telomere attrition can also lead to CIN via breakage-fusion-bridge cycles of chromosomes with shortened telomeres, which will lead to chromosome fragmentation and thus structural copy number changes [17]. Collectively, a variety of factors, including, but not limited to deregulated kinetochore-microtubule interactions, replication stress, and telomere erosion, contribute the diverse karyotypic alterations instigated by CIN [9,14–18].”
2.3
In 2.1-2.4, though the author described some consequence of CIN, the relationship of CIN and these malignant phenotypes such as EMT should be explained. For example, in 2.3, the author only indicated that CIN leads to a deregulation of metabolic signaling, but how?
Thank you for this comment. We agree that a clear explanation of the connection between Chromosomal Instability (CIN) and malignant phenotypes, such as the Epithelial-Mesenchymal Transition (EMT) was missing. In response to your suggestions:
2.3.1
We revised sections 2.1-2.4 to clarify how CIN can contribute to malignant transformation. We included a more detailed description of the downstream effects of CIN and how these effects can impact cellular behavior and lead to phenotypic changes.
“By promoting karyotype evolution, CIN acts as a catalyst to acquire malignant phenotypes, including EMT [1,2]. For instance, CIN was described to affect cellular pathways like TGF-β and WNT, and can change the tumour microenvironment to promote EMT, for example, through induction of hypoxia [3–5]. Additionally, CIN was found to inhibit the expression of the epithelial marker E-cadherin, thus promoting transition to a more mesemchymal phenotype [6].”
2.3.2
In section 2.3, we have elaborated on the relationship between CIN and the deregulation of metabolic signalling. We clarify how CIN can result in metabolic deregulation, supported by recent refences.
“How does CIN lead to this altered cellular metabolism? One important factor is that the resulting aneuploidy will affect copy numbers of metabolic enzymes and thus alter the expression of these genes, resulting in imbalances in various metabolic pathways [7,8]. Additionally, CIN-instigated activation of oncogenes like c-Myc or the loss of tumour suppressors such as p53 were found to reprogram metabolic dynamics [9,10]. By altering expression of genes that sense nutrients CIN can also impact responsiveness to these nutrients, while will likely change metabolic feedback mechanisms [11]. Furthermore, oxidative stress resulting from CIN can impair mitochondrial function, which will also affect cellular metabolism [12,13]. Finally, CIN may affect copy numbers of genes involved in growth factor signalling, such as insulin or IGF-1 signalling, which are important regulators of the cellular metabolism [14]. Together this explains the profound effect that CIN has on the cellular metabolism [15,16]. “
2.4
In 2.4.1, major change of TME cell subsets, either phenotype or function should be mentioned. For example, whether the MDSC accumulation, or CD8 T cell dysfunction correlates with CIN? Is critical chemokine that shaping TME influenced by CIN?
In response to your recommendation, we extended the discussion in section 2.4.1, which now describes the changes in TME cell subsets in relation to CIN. Specifically, we describe the accumulation of Myeloid-Derived Suppressor Cells (MDSCs) and the potential dysfunction of CD8 T cells in the context of CIN.
Furthermore, we now describe the influence of CIN on key chemokines that shape the TME, including the interplay between CIN and the chemokine landscape of the tumour milieu.
“Cancer cells that display CIN furthermore might promote recruitment of myeloid-derived suppressor cells (MDSCs) to the TME, possibly due to increased secretion of damage-associated molecular patterns and tumour-derived factors [17]. Furthermore, the chronic inflammation driven by CIN+ cancer cells may yield a more tumour suppressive environment, which in turn will contribute to the dysfunction of CD8 T cells [17–19]. Finally, the aneuploidies resulting from CIN can alter the expression of important chemokines, such as CCL5 or CXCL9/10, which will also impact the immune cell composition of the TME [20–22].”
2.5
The role of CIN in averting type I interferon signaling should be discussed.
In response to your recommendation we have included a new subsection in 2.1 dedicated to the interplay between CIN and type I interferon signalling. In this section, we discuss the mechanisms by which CIN might influence type I interferon signalling, either by direct modulation of key molecules or through indirect effects on the immune landscape.
“As such, the complex interaction between CIN-associated DNA leakage and the activation of cGAS-STING signalling emerges as an important driver of metastasis, with immune components within the TME playing a pivotal role in orchestrating this process and potentially powerful strategies to exploit the CIN-induced inflammatory response to treat cancer [23] (also see Figure 1).
In addition to activating cGAS-STING signalling, CIN also triggers type I interferon (IFN) signalling, a cornerstone of the antiviral defence and tumour immune surveillance [24] . Intriguingly, CIN+ cancers were found to circumvent immune surveillance through amplification of oncogenes, which yielded inhibition of the cGAS-STING pathway and to reduced production of type I IFN [17,24]. Additionally, CIN was found to disrupt IFN signalling by altering the expression of essential molecules such as STAT1 or IRF9 or the amplification of negative feedback mechanisms such as upregulation of SOCS [24,25]. Therefore, while the cytoplasmic DNA resulting from CIN might trigger cGAS-STING and IFN signalling to activate immune surveillance, CIN+ cancers quickly adapt by alleviating the immune activating IFN signalling to prevent immune clearance. Therefore, new therapies that reactivate immune recognition of CIN+ cancer cells are urgently needed.”
2.6
CIN is a Driver of Metastasis, the mechanism should be discussed
Thank you for your feedback, this is indeed an important point for which important progress has been made recently.
We therefore added a new section that explains how CIN promotes metastasis. This section discusses the cellular and molecular processes impacted by CIN that promote metastatic behavior in cancer cells. We explore how CIN can lead to genomic heterogeneity, allowing tumour cells to acquire invasive and migratory abilities. We discuss how CIN-driven changes can influence the tumour microenvironment, promoting an environment that facilitates metastatic spread.
“In summary, CIN plays a crucial role in promoting metastasis by affecting both the cellular and non-cellular components of the tumour microenvironment [23]. CIN is associated with an immunosuppressive environment, which is characterised by the presence of myeloid-derived suppressor cells and regulatory T cells that help cancer cells evade detection [17,18,26,27]. CIN also leads to changes in the ECM, making tumours more rigid and invasive [28,29]. Each of these effects along with CIN's ability to promote angiogenesis [30], have the potential to be exploited as therapeutic targets to treat cancers with a CIN phenotype (Figure 1).”
2.7
Sustained cGAS-STING activation can lead to therapeutic resistance. The relationship between CIN and therapeutic response and resistance should be mentioned
Thank you for pointing this out. While our paper primarily focuses on the mechanisms by which CIN contributes to metastasis in TNBC, we appreciate the relevance of therapeutic response and resistance. We therefore extended the section on how CIN promotes therapeutic resistance.
"While improved cancer therapies have significantly improved survival rates and quality of life in patients with various malignancies, therapy resistance is one of the most important factors affecting overall patient survival [31,32]. The emergence of drug resistance is often attributed to new mutations in tumour cells during treatment [33–35] . CIN is an important contributor to such drug resistance [36,37]. For instance, delays in the G1 phase of the cell cycle imposed by aneuploidy, the result of CIN, were described to play a role in increased resistance to chemotherapy drugs like cisplatin and paclitaxel [38]. In line with that, CIN is associated with higher recurrence rates and worse prognosis in neuroendocrine neoplasms [39,40] and a crucial indicator of metastasis and therapy resistance in cancers like hepatocellular carcinoma, colorectal cancer, and breast cancer, highlighting its potential as a target for treatment.”
2.8
Some sections should be re-arranged and refined, for example, the discussion should summarize the significance and the meaning of the review.
Thank you for your valuable feedback. We have added various sections to the review that hopefully improve the overall structure. Furthermore, we rewrote the conclusion to summarize the meaning of the review.
“In this review, we have explored the effects that CIN can have on the TME locally and at distant sites. As CIN discriminates cancer cells from non-cancer cells, it provides an attractive target for selective cancer therapy. CIN results in an accumulation of aneuploid cells with various karyotypes, thus promoting cancer cell evolution. As a dynamic factor, CIN contributes to the cancer progression, and is therefore associated with tumour recurrence, metastasis, and treatment resistance. On the other hand, CIN promotes inflammatory phenotyopes in cancer cells as a result of chronic cGAS-STING pathway activation, which ultimately leads to mobilization of immune cells to clear CIN+ cells. CIN+ cancer cells appear to circumvent this immune recognition through various mechanisms. Our growing understanding of this inflammatory response and mechanisms of immune suppression within the TME open up novel opportunities for therapeutic intervention, including immune modulation within the TME of CIN+ cancers to reactivate the immune response to clear CIN+cancer cells and inhibition of the inflammatory response resulting from CIN to kill CIN+ cancer cells. While these approaches still require further work, they do hold a great promise towards novel cancer therapies that target CIN+ cancers more effectively and with fewer side in the near future.”
Reviewer 3 Report
Comments and Suggestions for Authors
Siqi Zheng et al. reviewed the role of chromosomal instability in cancer progression. This topic is interesting, and the author give insight into multiple consequence of chromosomal instability, and provide some knowledge of finding potential therapeutic strategies. My major concern is it seem that the author mainly focuses on the results in the review, while the relationship with CIN and the mechanism underlying, should be emphasized in the manuscript
1. Although this manuscript is well prepared, the language should be polished by a native speaker at deep extent.
2. In the second part, the author should at least give a brief introduction and explanation of cause of chromosomal instability in cancer cells, such as karyotypic changes, chromosome mis segregation, and some great reviews could be referred to. (1)
3. In 2.1-2.4, though the author described some consequence of CIN, the relationship of CIN and these malignant phenotypes such as EMT should be explained. For example, in 2.3, the author only indicated that CIN leads to a deregulation of metabolic signaling, but how?
4. In 2.4.1, major change of TME cell subsets, either phonotype or function should be mentioned. For example, whether the MDSC accumulation, or CD8 T cell dysfunction correlates with CIN? Is critical chemokine that shaping TME influenced by CIN?
5. The role of CIN in averting type I interferon signaling should be discussed.
6. CIN is a Driver of Metastasis, the mechanism should be discussed
7. Sustained cGAS-STING activation can lead to therapeutic resistance. The relationship between CIN and therapeutic response and resistance should be mentioned
8. Some sections should be re-arranged and refined, for example, the discussion should summarize the significance and the meaning of the review.
Reference
1.Bakhoum S F, Cantley L C. The multifaceted role of chromosomal instability in cancer and its microenvironment[J]. Cell, 2018, 174(6): 1347-1360.
Comments on the Quality of English LanguageMinor revision needed.
Author Response
Reviewer 3
In the Review manuscript entitled "Chromosomal Instability-Driven Cancer Progression: Interplay with the Tumour Microenvironment and Therapeutic Strategies," Siqi Zheng, Erika Guerrero-Haughton, and Floris Foijer explore the influence of chromosomal instability on the tumor microenvironment and discuss potential therapeutic approaches for cancer treatment. The review is commendably well-written and organized. However, there are two major issues that need to be addressed:
3.1 and 3.2
Inappropriate terminology usage:
- a) The authors frequently use the term "chromosomal instability (CIN)" when "aneuploidy" or "heterogeneity" would be more appropriate. It's crucial to distinguish between these terms since CIN and aneuploidy represent distinct traits with potentially different clinical implications. The rates of random chromosomal aberrations (CIN) should not be conflated with the level of aneuploidy, as various factors contribute to the selection of chromosomal aberrations leading to clonal aneuploidy.
- b) At times, CIN is used as an umbrella term, encompassing the entire CIN phenotype (including aneuploidy, heterogeneity, instability, micronuclei, lagging chromosomes, multipolar mitoses, etc.), while in the next paragraph, it may be used to specifically refer to the instability process itself, measurable by rates of chromosomal instability. Clear delineation of terminology, along with an indication of how specific aspects of the CIN phenotype relate to characteristics of the tumor microenvironment and potential treatment strategies, would enhance the manuscript's clarity.
Thank you for bringing up these important points. In response to both of these points:
We have thoroughly reviewed the manuscript and made corrections where "CIN" was mistakenly used instead of "aneuploidy" or "heterogeneity." We now better differentiate between CIN, (i.e. rate of chromosomal aberrations), and aneuploidy (i.e. the presence of an abnormal number of chromosomes within a cell). Furthermore, we edited the text to use the term “CIN” more consistently. Importantly, we included a clear definition to distinguish between CIN, aneuploidy and heterogeneity at the beginning of the text.
"CIN refers to an increased frequency of errors in mitosis and can lead alterations of whole chromosome copy numbers as well as to structural rearrangements [41,42]. The result of CIN is aneuploidy, a state in which cells have an unbalanced DNA content. Furthermore, CIN will lead to variability between cancer cell karyotypes, a concept known as intratumour karyotype heterogeneity.”
Additionally, we provide specific examples and context to demonstrate how each aspect of the CIN phenotype impacts the tumour microenvironment. We more thoroughly discuss the potential implications for treatment strategies in a more organized and clear manner.
“For instance, delays in the G1 phase of the cell cycle imposed by aneuploidy, the result of CIN, were described to play a role in increased resistance to chemotherapy drugs like cisplatin and paclitaxel [38]. In line with that, CIN is associated with higher recurrence rates and worse prognosis in neuroendocrine neoplasms [39,40] and a crucial indicator of metastasis and therapy resistance in cancers like hepatocellular carcinoma, colorectal cancer, and breast cancer, highlighting its potential as a target for treatment[24,43–45].”
We hope that these changes will improve readability of the manuscript and provide the readers with a better understanding of the concepts of CIN, aneuploidy and heterogeneity.
3.3
c) The manuscript refers to "CIN rates" in relation to both cell lines and tumors (e.g., Lines 31, 42, 170...). Please explain how CIN rates were measured for tumor samples in the cited papers.
While this is a very important point, as many methods claim to measure CIN, but in fact measure aneuploidy, we choose to not explicitly mention how CIN (or aneuploidy) was measured for every paper that claims effects from CIN. The reason is that the methods use vary greatly between studies and that this would distract from the main message: what is the effect of CIN and resulting aneuploidy on the TME. We hope this is acceptable.
Line 31
Now Line 48
The chromosomal instability rate in tumours, including PitNETs, is measured here using array Comparative Genomic Hybridization (aCGH). This technique compares tumour DNA to a reference to quantify genomic imbalances, providing important information about the extent of copy number variations and their prognostic implications [27].
Reference:
- Lasolle, H.; Elsensohn, M.-H.; Wierinckx, A.; Alix, E.; Bonnefille, C.; Vasiljevic, A.; Cortet, C.; Decoudier, B.; Sturm, N.; Gaillard, S.; et al. Chromosomal Instability in the Prediction of Pituitary Neuroendocrine Tumors Prognosis. Acta Neuropathol. Commun. 2020, 8, 190, doi:10.1186/s40478-020-01067-5.
Line 42
Now Line 48
The rate of chromosomal instability can be determined by analyzing chromosomal variations and phenotypic changes in these clones. This can be done by observing differences in the expression of proteins such as E-cadherin, desmoplakin, desmoglein, and ZEB1, which are indicative of epithelial or mesenchymal characteristics associated with chromosomal instability [51].
Reference:
- Gao, C.; Su, Y.; Koeman, J.; Haak, E.; Dykema, K.; Essenberg, C.; Hudson, E.; Petillo, D.; Khoo, S.K.; Vande Woude, G.F. Chromosome Instability Drives Phenotypic Switching to Metastasis. Proc. Natl. Acad. Sci. U. S. A. 2016, 113, 14793–14798, doi:10.1073/pnas.1618215113.
Line 170
Researchers use techniques like fluorescence in situ hybridization (FISH) and chromosome painting to measure the rate of chromosomal instability in tumours. FISH uses specific probes to identify cytogenetic abnormalities such as chromosomal breaks, fragments, and end-to-end fusions, while chromosome painting quantifies chromosomal translocations. These methods enable the direct visualization and quantification of chromosomal abnormalities in tumour cells' metaphase spreads, providing a precise evaluation of genomic instability levels [75].
Reference:
- Samper, E.; Nicholls, D.G.; Melov, S. Mitochondrial Oxidative Stress Causes Chromosomal Instability of Mouse Embryonic Fibroblasts. Aging Cell 2003, 2, 277–285, doi:10.1046/j.1474-9728.2003.00062.x.
Line 170
Cytogenetic analysis is used to measure the rate of chromosomal instability in tumours by identifying and quantifying structural and numerical chromosomal aberrations. This involves evaluating the occurrence of various chromosomal abnormalities, such as premature centromeric separation, dicentrics, chromatid breaks, end-to-end fusions, iso-chromatid lesions, and numerical polyploidization. These abnormalities can be identified through changes in chromosome numbers and structure resulting from treatments like N-succinimidyl N-methylcarbamate. These observations are usually made by examining treated cells under a microscope and counting the frequency of these abnormalities [76].
Reference:
Reference:
- Mishra, P.K.; Raghuram, G. V; Panwar, H.; Jain, D.; Pandey, H.; Maudar, K.K. Mitochondrial Oxidative Stress Elicits Chromosomal Instability after Exposure to Isocyanates in Human Kidney Epithelial Cells. Free Radic. Res. 2009, 43, 718–728, doi:10.1080/10715760903037699.
Now Line 50
Researchers commonly analyze cellular and genetic markers to measure the rate of chromosomal instability in tumours. These markers include aneuploidy, complex karyotypes, lagging chromosomes, chromosome bridges, and micronuclei. These markers are quantified using techniques such as G-band karyotyping, confocal microscopy, and time-lapse micrography. These techniques were employed in a study on RanGAP1-depleted mouse embryonic fibroblasts (MEFs) [30].
Reference:
- Gong, Y.; Zou, S.; Deng, D.; Wang, L.; Hu, H.; Qiu, Z.; Wei, T.; Yang, P.; Zhou, J.; Zhang, Y.; et al. Loss of RanGAP1 Drives Chromosome Instability and Rapid Tumorigenesis ofOsteosarcoma. Dev. Cell 2023, 58, 192-210.e11, doi:10.1016/j.devcel.2022.12.012.
Now line 97
Researchers measure the chromosomal instability rate in tumours by assessing cellular and chromosomal anomalies, such as the fraction of metaphases with chromosomal aberrations and the percentage of anaphases with chromosomal bridges. These measurements are usually performed by examining and quantifying stained tumour cell samples under a microscope, as shown in the study on the impact of A3A on chromosomal instability in pancreatic ductal adenocarcinoma (PDA) cells [39].
Reference:
- Wörmann, S.M.; Zhang, A.; Thege, F.I.; Cowan, R.W.; Rupani, D.N.; Wang, R.; Manning, S.L.; Gates, C.; Wu, W.; Levin-Klein, R.; et al. APOBEC3A Drives Deaminase Domain-Independent Chromosomal Instability to PromotePancreatic Cancer Metastasis. Nat. cancer 2021, 2, 1338–1356, doi:10.1038/s43018-021-00268-8.
Now Line 156
To measure the chromosomal instability (CIN) rate in tumours, especially in colorectal cancer, it is important to identify widespread aneuploidy and loss of heterozygosity (LOH). Genomic analysis techniques such as karyotyping, fluorescence in situ hybridization (FISH), and comparative genomic hybridization (CGH) can be used to detect chromosomal number imbalances and structural abnormalities that indicate CIN [65].
Reference:
- Pino, M.S.; Chung, D.C. The Chromosomal Instability Pathway in Colon Cancer. Gastroenterology 2010, 138, 2059–2072, doi:10.1053/j.gastro.2009.12.065.
3.4
Inadequate citation relevance: The manuscript includes a total of 129 references, some of which are not directly relevant to the statements they are meant to support. In several instances, these references are placed after statements that are either unrelated or only tangentially related to the issues discussed in the cited papers. This issue persists throughout the manuscript and necessitates a thorough review of the cited papers by the authors to ensure that references are appropriately placed or replaced with more relevant ones.
Thank you for bringing up this point and our apologies for these inconsistencies. We have thoroughly reviewed all references to ensure they directly support the statements they are associated with. Irrelevant citations have been replaced with relevant ones, and tangential references have been removed or repositioned within the text.
3.5
Minor points:
Ln 14-15: sentence needs editing.
From
“CIN refers to an increased rate of changes in chromosome numbers or structural copy number alterations during cell division, thus promoting chromosome copy number alterations and unbalanced structural rearrangements.”
To Line 17
“CIN refers to an increased frequency of errors in mitosis and can lead alterations of whole chromosome copy numbers as well as to structural rearrangements.”
Ln 207-208: sentence needs editing.
From“This effect correlates with decreased levels of the ECM component EFEMP1 in these tumours and reinstating EFEMP1 levels enhances overall mitotic fidelity, suggesting that reversing ECM changes can decrease tumour cell aggressiveness by downregulating CIN rates [72].”
To Line 258
“This phenomenon correlates with reduced expression of EFEMP1, a constituent of the ECM [99]. Restoring EFEMP1 levels was found to improve mitotic fidelity, suggesting that modulating the ECM by restoring EFEMP1 levels might reduce CIN rates [46].”
Round 2
Reviewer 2 Report
Comments and Suggestions for Authors
The revised manuscript (MS) shows notable improvements, including the resolution of previous concerns regarding references. However, the authors persist in the occasional use of incorrect terminology related to chromosomal instability (CIN).
The authors have stated their choice to refrain from detailing the measurement methods of CIN or aneuploidy across various studies, to maintain focus on the main message concerning the impact of CIN and resulting aneuploidy on the tumor microenvironment (TME). While the intent to avoid distraction is understood, the continued misuse of the term “CIN” (particularly “CIN rates”) in place of more appropriate terms, remains problematic. This error, despite being common in the field, should be rectified to enhance the clarity and accuracy of this MS.
For example, in their response, the authors cite Pino and Chung (2010): “To measure the chromosomal instability (CIN) rate in tumors, especially in colorectal cancer, it is crucial to identify widespread aneuploidy and loss of heterozygosity (LOH). Genomic analysis techniques such as karyotyping, fluorescence in situ hybridization (FISH), and comparative genomic hybridization (CGH) can be used to detect chromosomal number imbalances and structural abnormalities that indicate CIN [65].”
Reference: 65. Pino, M.S.; Chung, D.C. The Chromosomal Instability Pathway in Colon Cancer. Gastroenterology 2010, 138, 2059–2072, doi:10.1053/j.gastro.2009.12.065.”
However, Pino and Chung do not discuss measuring chromosomal instability rates in tumors. They detail the methodological challenges in establishing criteria for a “CIN-positive” tumor.
The methodologies mentioned in the authors' response (CGH, aCGH, karyotyping, FISH, cytometry, LOH analysis, changes in expression) are indeed used to detect the presence of the CIN phenotype in tumors, and to measure the level of aneuploidy, or frequencies of rearrangements, but not the rates of chromosomal instability. To date, the rates of CIN have been measured only in cancer cell lines (as the number of changes in karyotype or genome per cell division).
The consistent substitution of the term “rates of CIN” with more appropriate terms like “level of aneuploidy,” “CIN phenotype,” or “frequency of chromosomal rearrangements”, etc., is necessary throughout the manuscript.
In conclusion, the manuscript would benefit significantly from the correct use of CIN-related terminology. With these minor yet critical amendments, the paper should be well-prepared for publication.
Comments on the Quality of English LanguageGood quality of English language
Author Response
Many thanks for bringing this point up once more: we apologise for not incorporating this critique sufficiently. We now only use CIN rates in cases where drugs/processes alter the CIN rates, or when CIN is quantified in cell lines (using timelapse imaging). In case of cancer samples, we replaced CIN with terms like 'aneuploidy landscapes, or aneuploidy rates and intratumoural karyotype heterogeneity as a proxy of CIN. We hope that by making these changes, we now better discriminate between CIN and aneupldoidy, particularly when CIN is inferred from aneuploidy landscapes in tumours.